# Identification of stable areas in unreferenced laser scans for automated geomorphometric monitoring

Daniel Wujanz[1], Michael Avian[2], Daniel Krueger[3], Frank Neitzel[1]

[1]Institute of Geodesy and Geoinformation Science, Technische Universität Berlin, Berlin, 10623, Germany
[2]Austrian Institute of Technology GmbH, Tulln, 3430, Austria
[3]GFaI - Society for the Advancement of Applied Computer Science, Berlin, 12489, Germany

*Correspondence to*: Daniel Wujanz (daniel.wujanz@tu-berlin.de)

**Abstract.** Current research questions in the field of geomorphology focus on the impact of climate change on several processes causing subsequently natural hazards. Geodetic deformation measurements are a suitable tool to document such geomorphic mechanisms e.g. by capturing a region of interest with terrestrial laser scanners which results in a so-called 3D point cloud. The main problem in deformation monitoring is the transformation of 3D point clouds captured at different points in time (epochs) into a stable reference coordinate system. In this contribution a surface-based registration methodology is applied, termed the iterative closest proximity algorithm (ICProx), that solely uses point cloud data as input, similar to the iterative closest point-algorithm (ICP). The aim of this study is to automatically classify deformations that occurred at a rock glacier, an ice glacier as well as in a rock fall area. For every case study two epochs were processed while the datasets notably differ in terms of geometric characteristics, distribution and magnitude of deformation. In summary, the ICProx-algorithm's classification accuracy is 70% on average in comparison to reference data.

## 1    Introduction

Monitoring surface changes in hazardous areas has been an important task of the geodetic community in the last decades. Due to the predicted increase of natural disasters (Anderson and Bausch 2006) this problem domain will hence gain importance in the future. Potential forms of geomorphic natural hazards are rock falls or slides, landslides as well as slow surface movements which may be triggered or enforced by heavy rainfall events. Apart from geomorphic changes the characteristics and geomorphic consequences of glacial retreat are a crucial part of scientific research (Avian et al. 2017). This interest can be justified by e.g. the function of glaciers as water reservoirs (Bradley et al. 2006) that act as water supply for agriculture and communities downstream (Tirado et al. 2010). A valuable tool for the quantitative analysis of the earth's surface is represented by geomorphometry. The most common forms of geomorphometric monitoring aim to quantify kinematic (Schwalbe et al. 2008), areal (Lambrecht and Kuhn 2007) and/or volumetric changes over time. A particularly appropriate measurement technique for this task is terrestrial laser scanning (TLS) due to its combination of high precision

and information density (Avian et al. 2009). In order to process the acquired TLS data, an appropriate deformation model has to be chosen (Heunecke and Welsch 2000). Deformation measurements require the repeated acquisition of a region of interest (ROI) at different points in time that are referred to as epochs. Based on these multi-temporal measurements geometric changes that occurred in between epochs can be identified. Therefore a stable reference frame is required that is represented by temporally stable points respectively their 3D-coordinates. In order to provide this vital condition, deformed areas within the ROI have to be identified. Stable points or areas within the ROI can then be used to determine transformation parameters. These parameters allow the transformation of a successively captured epoch into the coordinate system of the reference epoch.

## 1.1 The processing chain of deformation monitoring

For the determination of geomorphometric measures TLS can be seen as an established acquisition technique (Oppikofer et al. 2008), whose measurements yield in so-called point clouds, i.e. 3D Cartesian coordinates of the captured object in a local coordinate system with its origin in the centre of the laser scanner. In order to quantify geometric differences in different epochs it is recommended to carry out all steps of the following processing chain:

- viewpoint planning (Bechtold and Höfle 2016), (Wujanz et al. 2016a ),
- data acquisition at different points in time (van Veen et al. 2017),
- transformation of individual epochs into a stable reference frame,
- quantification of deformation (Jaboyedoff et al. 2012), (Lindenbergh and Pietrzyk 2015).

The most delicate step in this processing chain is the transformation of single epochs into a common reference coordinate system, which is also referred to as registration or matching of point clouds. Erroneous effects that occur in this step have an immediate and systematic impact onto the quantification of deformation. Thus, all conclusions that are drawn based on the generated results are falsified. The detectability of geometric changes itself is influenced by the precision of the applied laser scanner (Böhler et al. 2003), (Wujanz et al. 2017) as well as the chosen algorithm for the actual quantification of deformation, see e.g. (Cignoni et al. 1998), (Girardeau-Montaut et al. 2005) or (Lague et al. 2013). While random influences onto the results are inevitable, systematic effects, such as usage of uncalibrated sensors (Neitzel 2006), (Salo et al. 2008), (Al-Manasir and Lichti 2015) or the aforementioned problem when computing transformation parameters, have to be avoided at all costs that otherwise yield bogus deformation.

## 1.2 State of research

Up to now the transformation into a reference coordinate system has been preferably carried out by usage of artificial targets and/or monumented reference points. Several disadvantages can be associated to this strategy such as (i) the effort of distributing the targets in the area of interest, (ii) the required survey of the targets in order to determine their 3D-coordinates

in a reference system and (iii) the limited extent of the placed targets within the region of interest. In addition, the size of the targets must increase with growing range between scanner and ROI – for ultra-long-range scanners with ranges of up to some kilometres the targets would have to measure several metres. For built-in systems that monitor a ROI over longer periods of time, rotational deviations of the sensor's home position (Lichti and Licht 2006) that may occur in between epochs are usually refined by using artificial targets, see e.g. (Adams et al. 2013). For this, the assumption has to be made that the majority of targets remains geometrically stable over the course of the survey campaign – which might not be the case. A geodetic principle in the context of computing transformation parameters is that corresponding points should surround the ROI. If this is not the case, extrapolative effects occur quite likely which falsify the outcome. The reason why targets are still commonly used in practice despite its numerous drawbacks can be found in the fact that all well-established operational procedures for total stations are directly transferable to TLS. Numerous alternatives for referencing point clouds exist which can be classified as follows:

- registration based on radiometric features,
- direct georeferencing of terrestrial laser scanners,
- use of geometric information.

Point clouds captured by terrestrial scanners also contain radiometric information (Höfle and Pfeifer 2007), which is also referred to as intensity, in addition to the geometric content. Intensity values are assigned to individual points and are based on the strength of a reflected signal. If the topology of points within a dataset is known, this information can be used to convert a point cloud into an image where intensity values represent the brightness of individual pixels. By doing this, well-established techniques from the field of image matching can be applied to 3D-datasets. As a first step so called keypoints have to be extracted, which are distinct features within a local neighbourhood in terms of their grey scale values. After keypoint extraction, descriptors are used to establish correspondences between keypoints from different datasets (Lowe 1999). Based on this information, transformation parameters can be computed. Since the long-term stability of radiometric information captured by TLS has not yet been studied, this strategy is not considered in the following. The successful application over short periods of time has been demonstrated by Böhm and Becker (2007).

Instead of computing transformation parameters based on established correspondences, the relation between different local TLS coordinate systems can also be achieved by observing their location and orientation within a superior coordinate frame. For this, additional sensors are required while the general concept is well known, for instance in aerial photogrammetry (Cramer et al. 2000). Methods for direct georeferencing of TLS were initially just of scientific interest, e.g. (Paffenholz et al. 2010) while lately several commercial systems emerged (Riegl 2017), (Zoller + Fröhlich 2017). A significant drawback of direct georeferencing is the extension of a scanner's error budget (Soudarissanane 2016), due to the use of additional positioning- and orientation sensors such as GNSS-equipment or electronic compasses. With increasing scanning range the

impact onto the relative rotation between two point clouds also increases, which is a result of the limited accuracy of compasses.

A vital prerequisite for deformation measurement is the repeated acquisition of a region of interest at different epochs. Consequently the corresponding point clouds feature a large degree of overlap that can be used to transform both epochs into a common reference coordinate system. Apart from the aforementioned method of using radiometric features, geometric information can be used in form of geometric primitives, such as planes (Gielsdorf et al. 2008), (Previtali et al. 2014), (technet 2017). As a first step planar segments have to be extracted from the original point clouds. Then, identical planes are computed instead of matching single points such as in radiometric approaches. By using planes instead of points, the precision of the resulting transformation parameters notably increases. However, the approach relies on the existence of planar areas within a region of interest and is hence mostly applied in urban environments. The most popular registration method uses redundantly captured regions of two point clouds and is called the iterative closest point algorithm (ICP) as proposed by Besl and McKay (1992). A substantial advantage of the last strategy over the aforementioned ones is the actual use of the information present in the point cloud. A drawback of these surface based algorithms is their dependence to a sufficient pre-alignment of two datasets.

### 1.3    Motivation

A general aim in point cloud processing is to achieve a fully automatic workflow. Hence, the motivation arose to develop an automatic procedure for deformation monitoring that rules out potential biases provoked by different users. In addition, the desired solution should be versatile in terms of its input data. An algorithm that satisfies the latter aspect is the ICP-algorithm that however relies on a satisfactory pre-alignment which leaves a methodical gap to the ideal of full automation. In order to achieve this prerequisite, several strategies appear to be suitable such as direct georeferencing, manual pre-alignment or pre-alignment algorithms. The desired solution should not rely on additional sensors or any form of user-interaction. Hence, a two-tiered strategy was chosen that combines different algorithms for the pre- (Aiger et al. 2008) as well as the fine registration (Besl and McKay, 1992) of point clouds. It is well known that ICP-based approaches are sensitive against deformed regions, which yields in erroneous registrations (Wujanz 2012). Thus, it is of vital importance to reveal deformations within the captured datasets and to exclude these areas from the computation of transformation parameters. This aspect directly yields the following research questions:

- Is it possible to automatically reject deformed regions within unreferenced laser scans?
- Is the outcome biased by the geometric characteristics of the input as well as dependent to magnitude and distribution of deformations?

Kromer et al. (2017) approach the first research question by applying Fischler and Bolles' (1981) random sample consensus (RANSAC) in order to automatically exclude correspondences between two epochs which either arise as a consequence of

deformation or false matches. Since Kromer et al. (2017) process data which has been captured by a permanent monitoring system at high temporal resolution, comparably low amounts of deformation can be expected between successive epochs. Thus, the risk of potential falsification of the outcome during the registration step by contaminated data should also be rather low, which otherwise may describe an insuperable hurdle for the parameter dependent RANSAC. Since we assume to deal

with deformation of larger magnitude and extent an alternative solution is required that will be subject of the section 2.

This contribution demonstrates the automatic registration of multi-temporal point clouds as a vital prerequisite for the computation of geomorphometric measures. Therefore a prototypical implementation of the iterative closest proximity-algorithm (ICProx) that was integrated into the software Final Surface (GFaI 2017) was deployed. A brief description of the

algorithm will be given in section 2. Three regions of interest have been processed featuring different landforms such as ice glaciers, rock glaciers and a rock fall area. The study areas are described in detail in section 3 while the computed results are subject of section 4. Section 5 summarises the findings presented in this article and gives an outlook on future endeavours.

## 2    Methods

In order to avoid falsification of the deformation measurement process and consequently the analysis of deformation, it is

unavoidable to automatically identify deformed areas within point clouds and to exclude them from the computation of transformation parameters. For this task the ICProx-algorithm was deployed that will be briefly introduced in the following. The ICProx is based on a spatial decomposition of the original datasets and identifies deformation respectively stable areas by comparison of locally computed transformation parameters for individual segments (Wujanz et al. 2016b). Congruent respectively stable areas in terms of geometry are detected by a combinatorial approach termed the maximum subsample

method (Neitzel 2004, p. 109 ff) which is, in terms of robustness against outliers/deformation, more reliable and hence superior to robust estimators or RANSAC. A detailed description can be found in (Wujanz et al. 2016b). The algorithm consists of three essential phases:

- Phase 1: Local alignment of datasets.
- Phase 2: Identification of both stable and deformation areas.
- Phase 3: Computation of transformation parameters based on stable areas.

Figure 1 illustrates the workflow of the iterative closest proximity-algorithm (ICProx) including the aforementioned phases and all related processing steps.

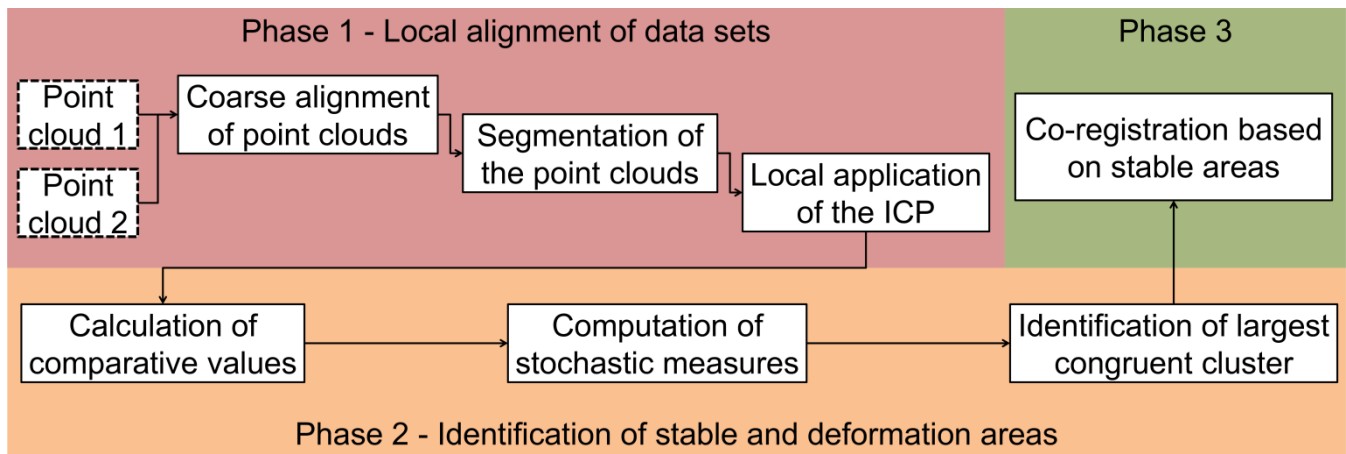

**Figure 1: Workflow of the iterative closest proximity-algorithm (ICProx)**

## 2.1 Local alignment of datasets

Phase 1 in turn contains three steps while the first one automatically carries out the pre-alignment of data. Therefore the
4PCS-algorithm as proposed by Aiger et al. (2008) is used in the first step. In the second step, the coarsely aligned data is
decomposed in cubes of equal size that are also referred to as octree cells (Samet 2006, p. 211 ff). As already shown in
Wujanz et al. (2016b) the choice of the octree cell size has notable effects onto the outcome. In general, a smaller octree size
leads to a higher degree of detecting stable regions while the computational effort notably increases. Hence, setting the
octree size is a balancing act between resolution and computation time. Subsequently the ICP-algorithm is locally applied
within each octree cell that represents the last step of Phase 1. Therefore so-called candidates are sampled in each cell of the
reference coordinate system. For all candidates in the reference point cloud, the ICP will determine corresponding points in
the second dataset. Thus, every cell receives an individual set of transformation parameters based on the established
correspondences.

## 2.2 Identification of stable and deformation areas

The general idea in decomposing the original data into subsets is to locally increase or decrease the portion of deformation in
order to identify geometric changes that occurred between scan acquisitions by means of suitable criteria in Phase 2.
Therefore the individual sets of transformation parameters are applied to the corresponding centroid of a cell. Subsequently
distances between centroids before and after the application of the computed transformation parameters are compared. This
strategy is in general comparable to the congruence analysis of geodetic networks by usage of combinatorial methods
(Neitzel 2004, p. 109 ff). Congruent and thus stable regions share the following characteristic which helps to identify them:
the distances between centroids before and after application of locally computed transformation parameters remain almost
identical. Another unique feature of the algorithm is the automatic computation of uncertainty measures (ISO 2008) that are
calculated for the local registration in every octree cell instead of using fixed thresholds. This information is required in

order to determine if an octree cell can be associated to a congruent set of cells or not. The general concept of this feature considers the fact that every scan of a stable object yields in different point sampling (Wujanz et al. 2016b). In Phase 3 the ICP is again applied to both epochs. This time however only stable regions, that were detected in Phase 2, serve as input which are processed as a whole and not in segments. The result of Phase 3 is a set of transformation parameters that forms the basis for the deformation estimation.

### 2.3    Some remarks on the spatial arrangement of deformation

Figure 2 depicts a test case on the left where stable areas are coloured in green while deformation is highlighted in yellow. The aim of the ICProx-algorithm is to identify these yellow areas and to exclude them from the final registration. The centre of the figure illustrates the decomposition of the input data. An individual set of transformation parameters is assigned to every cube by local application of the ICP-algorithm. After congruence analysis all segments are merged to one continuous dataset as depicted by green areas. Red regions have been classified as deformation and are thus excluded from the registration process.

From a geodetic perspective the most desirable arrangement would be a more or less connected region that is subject to deformation which is surrounded by stable areas. In addition, the ratio between stable and deformed regions should be rather large in order to being able to compute transformation parameters with the highest possible redundancy. However, the distribution of stable areas as well as their relative amount are usually unpredictable in practice since every region of interest has got its own individual characteristics. Countermeasures in order to receive favourable arrangements can be achieved by carrying out the surveys more frequently and/or to perform panorama scans which increases the chance of capturing additional stable areas. It is obvious that the result depicted on the right of Figure 2 may yield in imprecise transformation parameters due to the fact that a large amount of octree cells are subject to deformation while the arrangement of stable cells is unfavourable.

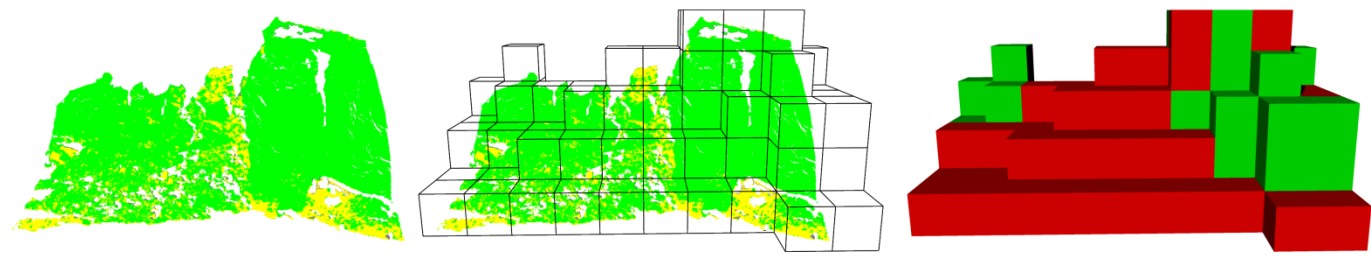

**Figure 2: (Left) Binary change map of a test case. (Centre) Octree structure that contains the scene. (Right) Stable (green cubes) and unstable (red cubes) octree cells after congruence analysis.**

## 3    Study areas

This study comprises three study areas in the Hohe Tauern area of the Central Alps, Austria, see Figure 3, monitoring three distinct geomorphic process groups:

- Hinteres Langtalkar: Rock glacier movement representing permafrost dynamics.
- Pasterze Glacier: Glacier retreat with subsequent proglacial processes.
- Mittlerer Burgstall: Rock fall activity.

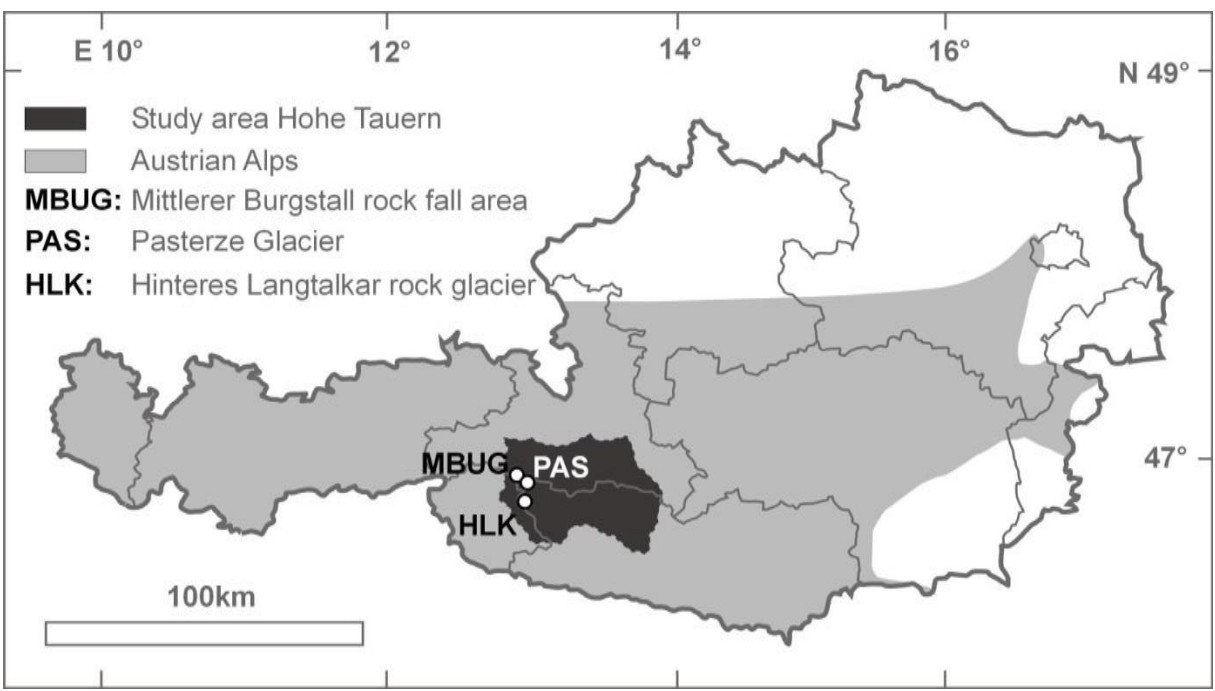

**Figure 3: Study areas Pasterze Glacier, Hinteres Langtalkar rock glacier and Mitllerer Burgstall rock fall area within Austria.**

All data sets were captured using a Riegl LMS Z620 laser scanner that applies a wavelength of 1550 nm and is capable to measure distances of up to 2000 m, in applications in the study area 1500 m were achieved. A summary of the characteristics of all epochs is given in Table 1.

**Table 1: Summary of all used measurements in every study area comprising date of acquisition, number of points: points per epoch, ground sampling distance (GSD)**

| Year | Date of acquisition | # of points [Mio.] | Ground sampling distance (GSD) [cm] | Scanning increment [deg] |
|------|--------------------|--------------------|-------------------------------------|--------------------------|
| Hinteres Langtalkar rock glacier | | | | |
| 2011 | 24.08.2011 | 1.6 | 6.6 – 41.8 | 0.040 |
| 2012 | 05.09.2012 | 7.8 | 4.8 – 33.1 | 0.029 |

| Pasterze glacier | | | | |
|------|------------|-----|---------|-------|
| 2010 | 13.09.2010 | 2.5 | 20 – 54 | 0.029 |
| 2012 | 10.09.2012 | 3.6 | 16 – 42 | 0.023 |
| Burgstall rock fall area | | | | |
| 2014 | 08.09.2014 | 2.3 | 20 – 54 | 0.029 |
| 2015 | 12.09.2015 | 1.6 | 20 – 54 | 0.029 |

## 3.1 The rock glacier Hinteres Langtalkar, Austria

The rock glacier Hinteres Langtalkar (N 46° 59', E 12° 47') is located in the Schober Mountains at an elevation between 2455 - 2720 m a.s.l. and is ~900 m long and ~350 m wide. Using TLS the rock glacier has been monitored since the year 2000 (Bauer et al., 2003). Extraordinary changes in surface elevation were detected around 2000 (Kaufmann and Ladstädter, 2002, Avian et al. 2005), leading to the assumption of acceleration most likely in 1994 (Avian et al., 2005). In order to geometrically quantify these surface elevation changes, a geodetic network was established in 1998 which has been surveyed since annually (Kienast and Kaufmann, 2004). The rock glacier itself shows distinct patterns of surface elevation changes and surface displacements. In addition, the rock glacier has reported to have a slower upper part and a rapidly moving lower part with maximum horizontal displacements of up to 2.5 m per year (Avian et al., 2009). The mean horizontal surface displacement rates vary between 0.10 - 0.18 m per year while maximum changes amount up to 3.6 m per year (Kaufmann and Ladstädter, 2009). Delaloye et al. (2008) estimate this rock glacier to be one of the currently fastest moving one in Europe which underlines its importance for the scientific community.

Despite the multiple use of TLS for monitoring geomorphological processes, see Oppikofer et al. (2008), TLS was rarely used for analysis of periglacial effects such as rock glacier movement patterns until the end of the last decade. First annual TLS observations were undertaken by Avian et al. (2005, 2009), or Bodin et al. (2008). Data acquisition was carried out from one of two known scanning positions (HLK, 2454.709 m a.s.l. on the left in Figure 4). The lowest and largest ground sampling distance (GSD) for the 2011 campaign summed up to 6.6 cm respectively 41.8 cm while the according measures for the survey of 2013 are 4.8 cm and 33.1 cm, see Table 1. These GSD values were computed based on the closest and furthest distance between scanner and scene. In Figure 4 all applied reference points are highlighted by white circles, the scanner position by a red ellipse with a white edge. These reference points were surveyed by geodetic means and are hence known in a superior coordinate system. It can notably be seen that the distribution is rather unfavourable from a geodetic point of view as the active zone of the glacier, that is highlighted by a white dashed line, is not surrounded by reference points. This may lead to extrapolative respectively leverage effects. In order to provide trustable reference transformation parameters a surface based registration has been conducted after deformed regions were manually excluded. The scanning sector is marked by red dashed lines. The shown picture was taken in 2012.

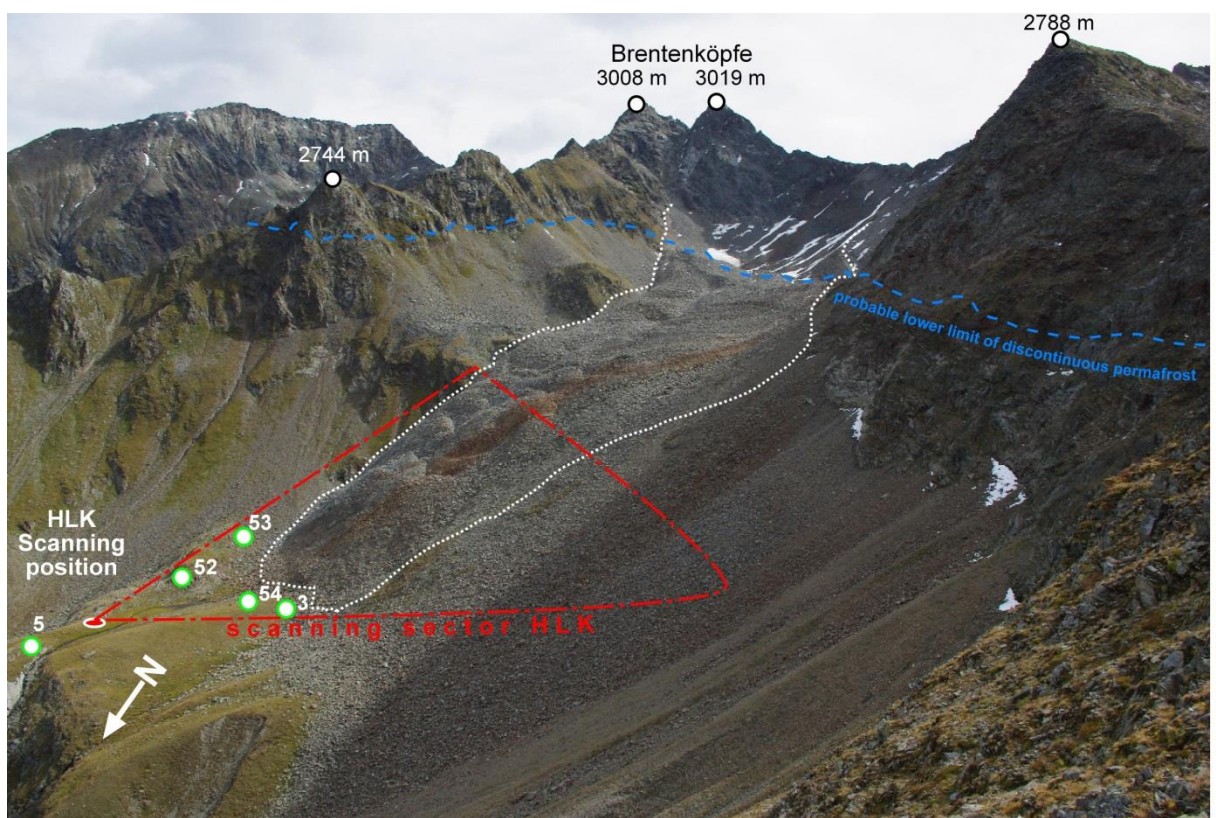

**Figure 4: Scanner configuration at Hinteres Langtalkar rock glacier. The scanning position HLK is indicated by a red ellipse with a white edge, the rock glacier outline by a white dashed line. White circles with green edges highlight reference points.**

## 3.2    The Pasterze glacier, Austria

The Pasterze glacier (N 47° 04', E 12° 44') is located in the central part of the Austrian Alps. The geometric behaviour of this glacier is of scientific interest since 1878 (Wakonigg and Lieb, 1996), which makes Pasterze glacier to one of the longest documented glaciers in the world. The glacier reached its latest maximum around 1852/56 in the Little Ice Age (Nicolussi and Patzelt, 2000) with a maximum area of 26.5 km². Since then the glacier constantly lost substance leading to a recent area of approximately 17.3 km² in 2009 (Kaufmann et al., 2015). Accordingly since the end of the Little Ice Age period Pasterze glacier lost approximately 35% of its area and about 60% of its volume. The current retreat of the terminus area of Pasterze glacier has been observed using TLS since 2001 covering an area of approximately 0.9 km². On average the annual elevation change within the debris covered part sums up to approximately 3.7 m while 6.3 m vertical loss was detected in the clean ice section between 2011 and 2012. A comprehensive view on the geomorphic consequences of the fast glacier retreat can be found in Avian et al. (2017). The scanning configuration at the Pasterze Glacier terminus area is depicted in Figure 5 while the picture was taken in 2008.

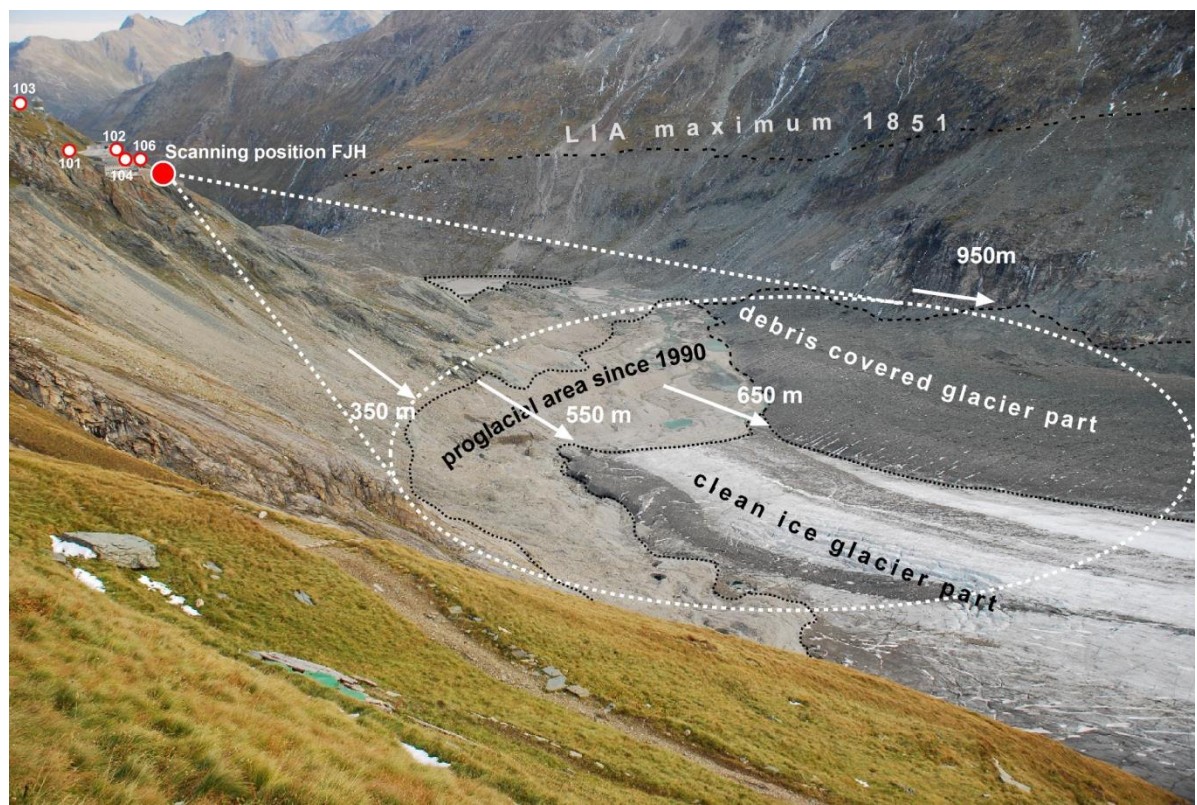

**Figure 5: Scanning configuration at Pasterze Glacier terminus area from the scanning position Franz-Josefs-Höhe (FJH). Dots near the scanning position represent the location of the reference points. Numbers near the arrows indicate the distance from the scanning position.**

The view at scanning position Franz Josefs Höhe (FJH, 2362.621 m a.s.l.) is excellent for geomorphic applications whereas the availability of stable areas lead to an unfavourable distribution of reference points that do not surround the ROI. As a consequence the computed transformation parameters based on the reference points can only be seen as an approximate solution as they can very likely cause extrapolative effects and hence have been refined by surface based registration. The necessity of revealing deformed regions prior to performing co-registration has already been extensively stated so that the ICProx-algorithm is deployed in the following to solve this task for a comparison among the datasets from 2010 and 2012. A summary of quality parameters of TLS measurements at Pasterze Glacier terminus area is given in Avian et al. (2017).

### 3.3    The Mittlerer Burgstall rock fall area, Austria

The rock fall area Burgstall comprises the two former nunatak mountains (Kellerer-Pirklbauer et al. 2012) of Mittlerer and Hoher Burgstall which were encompassed from Pasterze glacier until the beginning of the 20[th] century. The Mittlerer Burgstall (N 47° 06'and E 12° 42', 2933 m a.s.l.) shows a distinct NW-SE trending ridge forming two rock faces which were exposed by glacier shrinkage in the last 100 years. In early summer 2007 a large rock fall at the top of the ridge released

approximately 430000 m³ (V. Kaufmann, Graz University of Technology, personal communication) from a detachment area of approximately 3100 m² transporting debris to both sides of the ridge. Results from permafrost distribution modelling and ground temperature monitoring indicate (Kellerer-Pirklbauer et al., 2012) that the detachment area is located within or near the lower limit of permafrost. Consequently in 2010 a geodetic network consisting of one main scanning position (BUG, 2794.958 m a.s.l.) and six reference points was established. From every reference point each point can be viewed so that several scanning positions can be chosen to vary scanning geometry. The scanning configuration at scanning position Burgstall (BUG) can be seen in Figure 6. Yet again, the sampling of reference points has to be rated as unfavourable. Hence, initial transformation parameters have been computed based on the reference points. Afterwards, deformed regions were manually excluded while the remaining points were used to refine the relative alignment between point clouds based on the ICP.

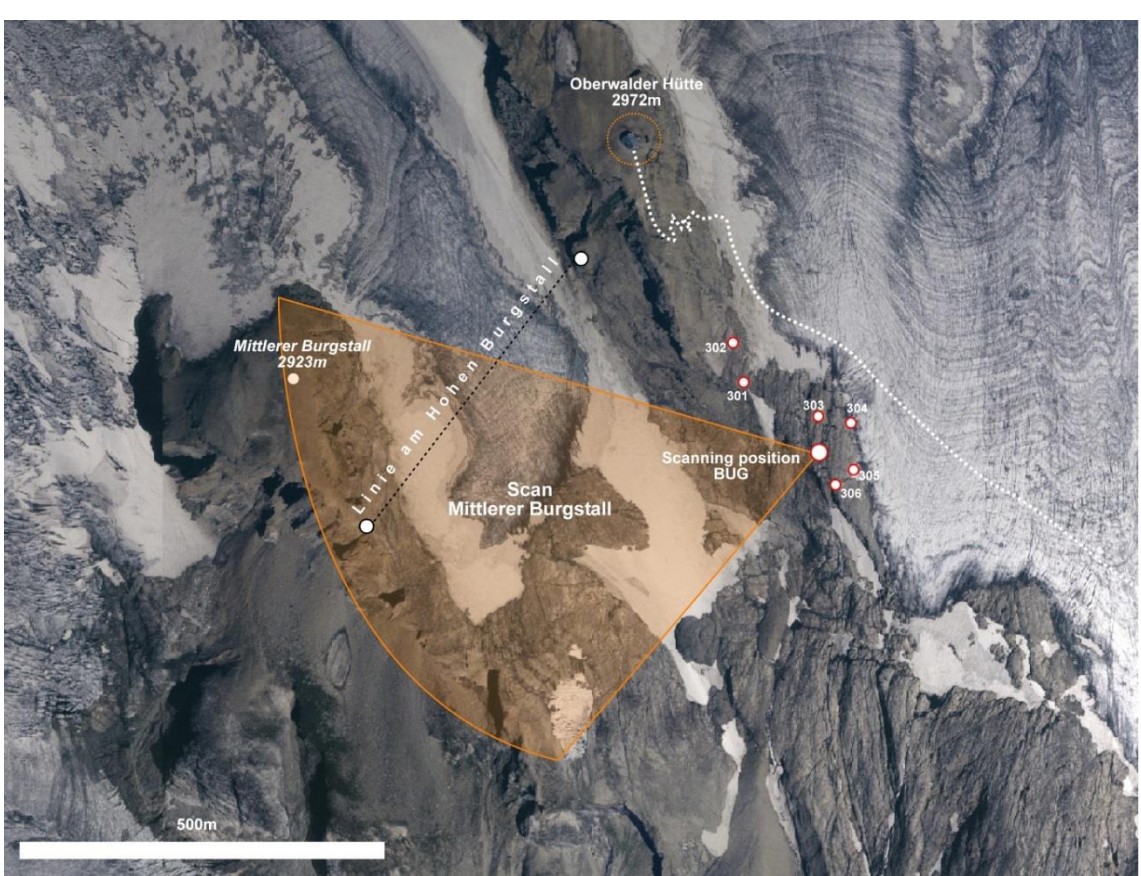

**Figure 6: Configuration at scanning position Burgstall ("Standpunkt BUG"). BUG is serving two objects, Mittlerer Burgstall and Hoher Burgstall rock fall area. Codes: 301 – 306: reference points; "Linie am Hohen Burgstall" cross section of traditional glacier surveying. Ortho image from 1998, Nationalpark Hohe Tauern©.**

## 4    Results and discussion

Every algorithm is sensitive to chosen settings as well as to the individual characteristics of the given data. In this section the three aforementioned regions of interest have been analysed by using the ICProx-algorithm. The input for the ICProx-algorithm consisted of two unreferenced point clouds per ROI given in different local scanner coordinate systems. Pre-alignment of the epochs was carried out by the 4-PCS-algorithm while subsequently the ICProx-algorithm was deployed in order to reveal regions that were subject of deformation. The inherent challenges of the datasets presented in section 3.1 and 3.2 can be associated to their sheer extent. A somewhat different degree of difficulty can be assigned to the ROI that was subject of section 3.3. This scene is demanding due to the scattered distribution of deformation and the comparably small magnitude of geometric changes as a consequence of processes after a huge rock fall.

Since the task of the ICProx-algorithm is essentially a classification problem – identify stable or deformed regions within a point cloud – the generated outcome can be numerically evaluated in form of confusion matrices while a visual inspection is carried out by so-called confusion maps. The reference data for the evaluation process is established by point clouds where deformed areas were manually removed, while the relative alignment of the remaining points was refined by surface based registration. Based on this data a point cloud was tinted in two colours representing the two classes of interest, namely deformation and stable region. The comparison between reference and the results computed by the ICProx-algorithm finally yields in a confusion matrix respectively confusion maps as depicted for instance on the left of Figure 7. Therefore the following colour scheme will be used and contains four elements, namely:

- Blue: Area is stable, detected as stable (true positive).
- Cyan: Area is stable, detected as unstable (false negative).
- Red: Area is unstable, detected as stable (false positive).
- Orange: Area is unstable, detected as unstable (true negative).

Note that a perfect result would only contain the first and last colour in the list. Yet, this aim is not achievable due to the applied octree structure.

### 4.1    Rock glacier Hinteres Langtalkar

After pre-alignment the entire dataset was decomposed into octree cells of 30 m so that in total 661 cells emerged. In order to locally determine transformation parameters candidates were sampled with 0.75 m on average. After congruence analysis, the largest subset contains 441 cells that were considered as being stable. Based on these areas an ICP-based registration was carried out leading to average residuals of 15.3 cm between the two point clouds. Afterwards the computed results were compared to the reference yielding in a confusion matrix as well as a colour coded confusion map which is depicted in Figure 7. The left part of Figure 7 shows octree cells where orange represents the rock glacier and red cells are considered as

being stable. This effect occurred predominately in the transition region between glacier tongue and surrounding areas. The reason why these cells have been falsely assigned to the stable class can be found in the very low magnitude of deformation in the transition zone of the rock glacier which has a similar scale as the uncertainty measures (ISO 2008). As a consequence, deformations of this size fall below the margin of detectability. The narrow red band on the left side of the scene was falsely assigned as being stable due to its limited spatial expansion.

The remaining parts of Figure 7 show change maps based on comparisons between the reference dataset and their according versions which were altered in position and orientation before application of the ICProx (centre) and ICP (right). Hence, the motivation for conducting this analysis is to reveal differences between the two surface based approaches against the reference data. A look at the centre part of Figure 7 shows that the majority of the scene complies within $\pm$ 5 cm while some parts feature deviations between 5 and 14 cm. These differences may be triggered by leverage effects due to an insufficient sampling of artificial targets for geo-referencing. This suspicion substantiates if one has a look at the centre part of Figure 4 where it can be seen that all targets do not surround the glacier tongue and are only located in north-westerly direction (above the glacier tongue in the figure). On average the differences between reference and the result computed based on the ICProx sum up to (i) 5.8 cm for points that are located above the reference surface and (ii) 4.5 cm for points below it while the standard deviation among the two datasets measures 5.7 cm. The right part of the figure again confirms that the ICP is not an appropriate algorithm for registration of point clouds that were subject of deformation. Again the same statistical measures were computed as before where the mean deviation among the datasets sum up to 17.4 cm for points above the reference respectively 23.9 cm for points below the reference surface. The according standard deviation of all residuals is 24.2 cm. It has to be mentioned that the listed statistical measures are somewhat whitewashed due to the fact that only deviations within the specified range between $\pm$ 50 cm were considered but not the grey coloured areas where the discrepancies exceed these limits.

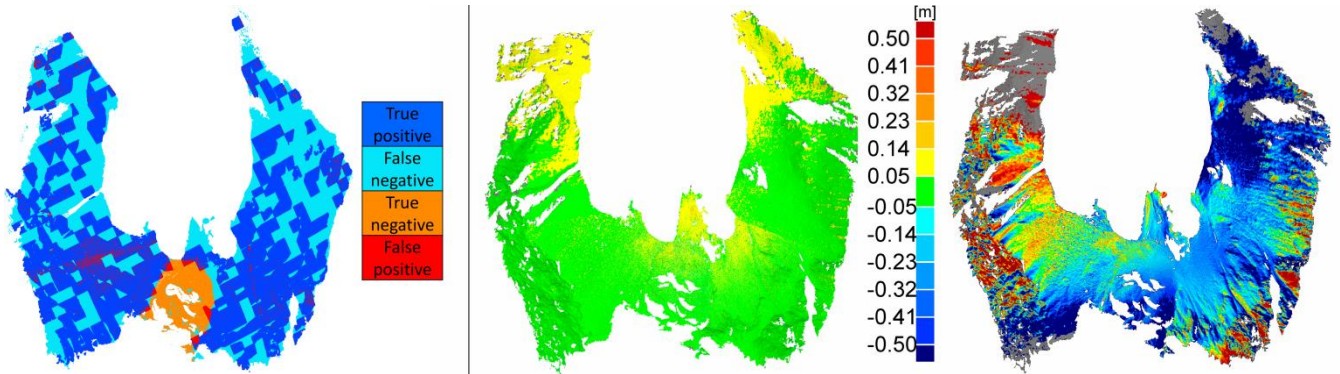

**Figure 7: Colour coded confusion map (left), change map based on a comparison to the reference data outcome by usage of the ICProx (centre) and the ICP (right) at Hinterer Langtalkar rock glacier.**

The numerical assessment of the generated results is given in Table 2 and reveals that a comparably large part of stable areas was not correctly assigned as indicated by the numbers in the cyan coloured cell. The reason for this can be found in the

weak geometric characteristics in the according octree cells. The overall classification accuracy sums up to a moderate 60% = 55% + 5% and can mostly be justified by the aforementioned geometric characteristics of the dataset. For the detection rate of stable areas a value of 60% = 55% / (55% + 37%) was achieved. This result can be rated in two ways. On the one hand it can be seen as a satisfactory result while on the other hand a substantial amount of valid information was not used. The

corresponding detection rate of deformation is 62.5% = 5% / (5% + 3%). After analysing the numbers in Table 2 it appears that the allegedly "simple" dataset with only 8% of deformation has turned into a challenging task for the algorithm. However, a second look at Figure 7 reveals that the stable areas surround the major area of deformation. Hence, a stable set of transformation parameters can be expected without extrapolative effects. The influence of 3% of false positive cells can be expected to be little due to the fact that the magnitude of deformation is comparably low in these parts and due to the low

ratio between true positive (55%) and false positive (3%) areas.

**Table 2: Confusion matrix for the classification of the rock glacier**

|  | Area is stable | Deformation |
|---|---|---|
| Classification: Area is stable | 55% | 3% |
| Classification: Deformation | 37% | 5% |

The change map based on the results of the ICProx-algorithm is illustrated on the left of Figure 8 and describes the basis for

deformation analysis by Geologists. In the right half of the figure the uncertainty values computed by the ICProx-algorithm for individual octree cells (Wujanz et al. 2016b) can be seen that form the statistic basis to distinguish stable areas and deformation. Hence, small uncertainty values allow detecting deformations of small magnitude. It is obvious that the distribution of uncertainty values is quite heterogeneous ranging from several centimetres to several decimetres. It can also be seen that the upper parts of the rock glacier have larger uncertainties. This can be explained by the fact, that the local

sampling rate is lower compared to the regions right below. The uncertainty values on the glacier tongue are also notably increased when compared to its surroundings. This is caused by the unsteady geometric characteristics of the glacier tongue in combination to the scanning configuration, as depicted in Figure 4.

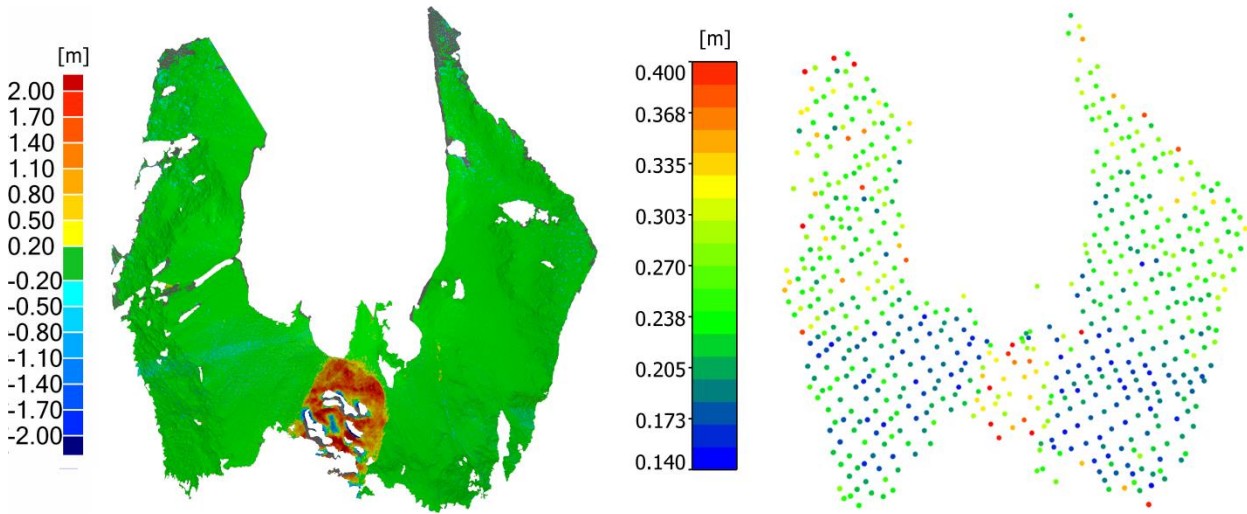

**Figure 8: Change map of the Hinterer Langtalkar (left) and spatial uncertainties [m] for individual octree cells (right).**

## 4.2    Pasterze glacier terminus area

For this ROI it can be expected that deformation occurred predominantly directly at the glacier's tongue so that comparatively large octree cells were defined. Hence, the size of the cells was empirically set to 52.5 m leading to 923 segments. On average candidates were sampled every 6 m which lead to approximately 40700 candidates. After application of the algorithm a congruent set consisting of 526 octree cells emerged that was finally used to compute the transformation parameters for deformation monitoring. The residuals after registration based on stable regions were 28.8 cm on average. An

explanation for this rather large quality measure can be found in the comparably low ground sampling distance in relation to the rugged terrain. Based on expert knowledge a set of reference transformation parameters was computed featuring undoubtedly stable areas. This procedure is very time consuming, yet crucial in order to avoid falsification of the reference dataset by deformation.

After generating the aforementioned reference dataset, quality assessment was carried out. Figure 9 depicts the confusion map on the left which was coloured according to the colour scheme proposed at the beginning of this section. In the centre of the left part the glacier tongue is highlighted in orange corresponding to large detected geometric changes. On the sides stable areas can be found which have been predominantly detected by the algorithm. Cyan coloured areas can be seen predominantly on the left of the tongue. These regions have not been detected as being stable due to the fact that their local

geometric characteristics did not feature notable topographic changes. Consequently the locally applied ICP converged into local minima, as the ICP requires a certain geometric contrast within the datasets. Red areas however have to be rated as being critical for the desired results. These regions can be found primarily in transition areas between glacier terminus and

adjacent stable parts of the point cloud. The reason why these areas were falsely classified can be explained by their magnitude of deformation – they fall below the margin of detectability. Single red spots in the otherwise stable flanks left and right of the tongue were triggered by single sliding or mass wasting processes (Avian et al., 2017). These appearances also fall below the margin of detectability, yet not due to their magnitudes but their limited spatial extent. On the right of the figure spatial uncertainty measures for each octree cell are shown which have been computed by the ICProx-algorithm. A heterogeneous distribution is apparent while an increasing characteristic can be seen towards the lower part of the image due to insufficient local sampling.

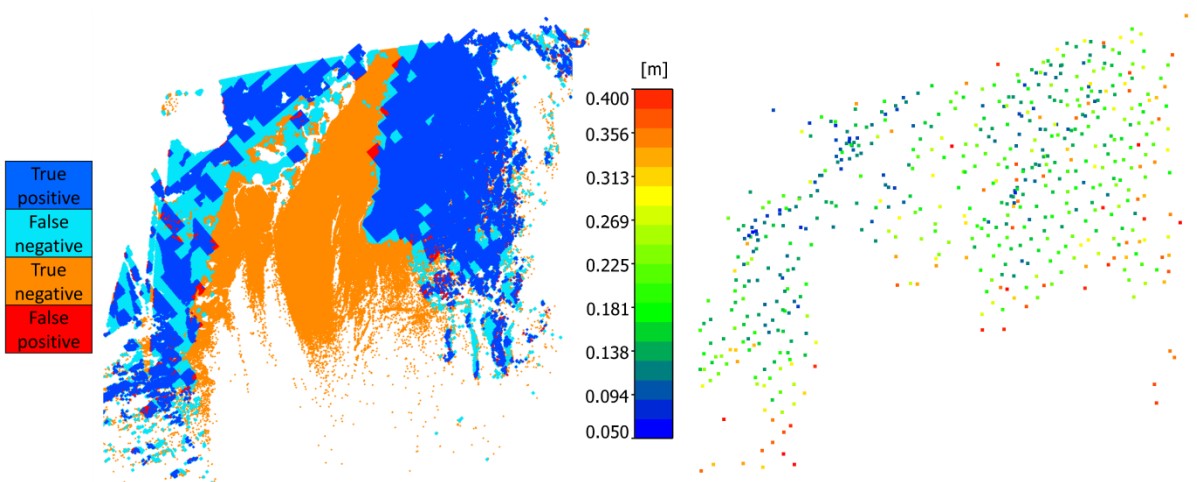

**Figure 9: Confusion map (left) and spatial uncertainties for individual octree cells (right).**

The numerical interpretation of the results can be found in Table 3. In total 19% of the dataset were falsely classified as deformation despite the fact that they actually remained geometrically stable. This effect does not provoke any substantial disadvantages, yet does simply mean that stable areas remain unused for the final registration. The portion of the critically false positive class, which is coloured in red in all tables and figures, sums up to 2%. These regions have undeniably an influence onto the registration yet can be rated as uncritical due to their comparably low magnitudes and low fraction in relation to the stable areas.

**Table 3: Confusion matrix for the classification of the ice glacier**

|  | Area is stable | Deformation |
|---|---|---|
| Classification: Area is stable | 43% | 2% |
| Classification: Deformation | 19% | 36% |

All in all the classification has to be rated quite positively with an overall accuracy of 79% = 43% + 36%. The detection rate for stable areas sums up to 69% = 43% / (43% + 19%), the one for deformation is even 95% = 36% / (36% + 2%). Consequently the algorithm can be characterised as being "careful" – in case of doubt areas are rather classified as being deformed.

Figure 10 features a point cloud that is coloured according to geometrical differences that occurred in between epochs. Large differences are notably visible at the glacier terminus area. Two regions need to be highlighted in particular. At first the orange to red coloured part has to be brought to attention vertical surface elevation changes in the magnitude of -6.5 to -10 m/a. In this part of the glacier the terminus area is only sparsely covered by debris. The second notable part is coloured in shades of yellow to green where the glacier tongue is covered by morainic debris. As a consequence the ablation rates are significantly lower compared to the clean ice part with vertical surface elevation changes in the magnitude of - 4 to -6 m/a (Kellerer-Pirklbauer et al. 2008).

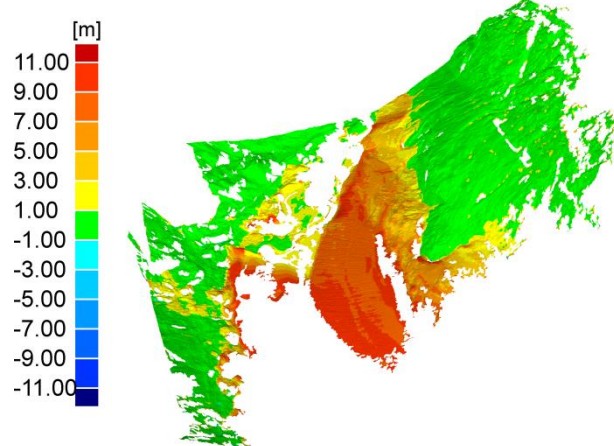

**Figure 10: Resulting change map based on the detected stable regions.**

### 4.3    Mittlerer Burgstall rock fall area

Just as before the input for the ICProx-algorithm consisted of two unreferenced point clouds, given in different local scanner coordinate systems. In order to decompose the datasets, an octree size of 5 m in all cardinal directions was chosen yielding in 3494 cells. Candidates were sampled with an average spacing of 0.5 m so that about 120000 correspondences had to be iteratively found by the ICP. After application of the ICProx-algorithm a congruent set of 2179 cells arose that was deployed to compute the desired transformation parameters for deformation monitoring. The residuals after registration based on stable regions were 17.2 cm on average. Again the explanation for this rather large quality measure can be found in the

comparably low ground sampling distance in relation to the rugged terrain. As before expert knowledge was used to create a set of reference transformation parameters by manually selecting undoubtedly stable areas. Based on this information a binary change map with the two classes stable area and deformation was created. This data served as a basis for the confusion map that is depicted on the left of Figure 11. In contrast to the two previously shown examples, several red areas are notable in which deformed regions were falsely classified as being stable. A close look at the red speckles which are located in the major rock fall area highlighted in orange, shows that most of them contain some locally restricted blue regions. As a result, a mixture between stable and predominantly deformed areas occurs within the corresponding octree cells and hence lowers the algorithm's margin of detectability. The reverse case can be seen on the right of the figure that is mostly tinted in blue where small red spots are notable. Another reason for the comparably frequent occurrence of red regions is the low magnitude of deformations. The right part of the figure shows spatial uncertainties for individual octree cells given in metres. A heterogeneous distribution of uncertainty values is apparent yet its variation appears to be weaker when compared to the other two examples. Note that the spatial uncertainties in large parts of the deformed region fall into the decimetre range.

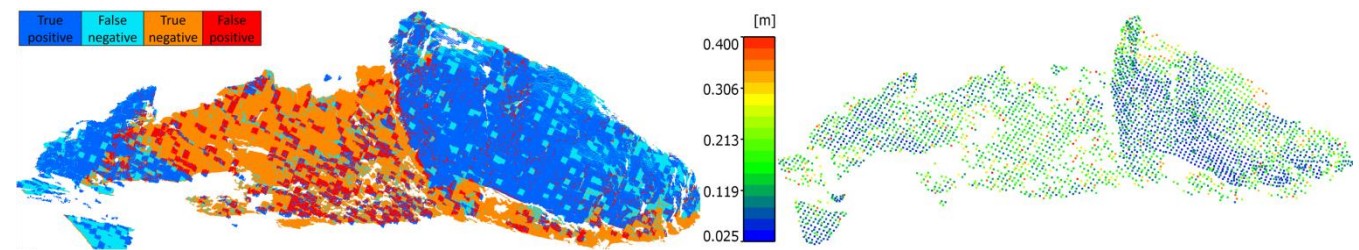

**Figure 11: Confusion map of the Burgstall dataset based on the classification with the ICProx (left) and spatial uncertainties for individual octree cells (right).**

The numerical assessment of the generated results is given in Table 4 and shows that a comparably large part of deformed areas was not correctly assigned as indicated by the numbers in the red coloured cell. The reason for this is twofold and can be found both in the ratio among the low magnitude of deformation and the corresponding local spatial uncertainties as well as the heterogeneous occurrence of geometric changes. The overall classification accuracy sums up to 72% = 43% + 29%. For the detection rate of stable regions a value of 74% = 43% / (43% + 15%) was achieved that marks the best value in this contribution. However, the corresponding detection rate of deformation only sums up to 69% = 29% / (29% + 13%).

**Table 4: Confusion matrix for the classification of the rock fall area**

|  | Area is stable | Deformation |
|---|---|---|
| Classification: Area is stable | 43% | 13% |
| Classification: Deformation | 15% | 29% |

Based on the reference transformation parameters a change map was generated that is depicted in Figure 12. The majority of geometric changes occurred in the centre of the figure as well as on the bottom of the right part. Apart from larger connected regions, local nests of rock fall can be seen especially on the upper right part of the illustration.

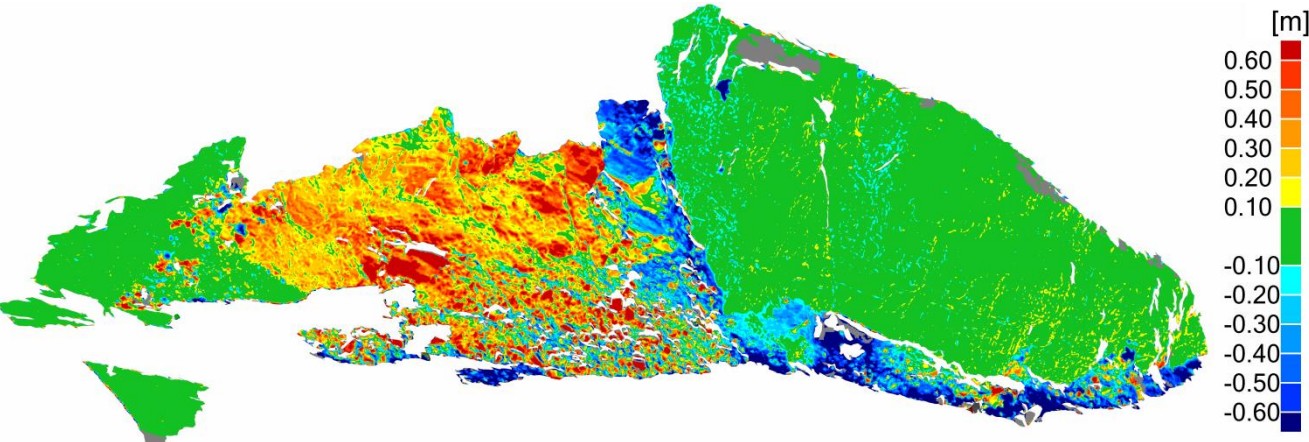

**Figure 12: Change map based on the detected stable regions – comparison between the datasets from 2014 and 2015.**

A look at Table 4 shows that 13% of all points were assigned to the false positive class. Hence, its influence onto the outcome is of interest in the following. Therefore the computed set of transformation parameters based on the detected stable regions was applied to the second epoch and compared to the generated reference dataset. On average the absolute deviation between reference and computed result is 7.37 cm. The corresponding change map in Figure 13 illustrates the differences in metres. In summary the processed dataset proved to be challenging and revealed boundaries for the ICProx-algorithm. Apart from magnitude and spatial distribution of deformation, the inherent relationship between detectability of geometric changes and local point density had a critical influence onto the outcome, that finally lead to the comparably large amount of falsely classified stable regions. Thus, future surveys in this region should be carried out at higher spatial sampling rates.

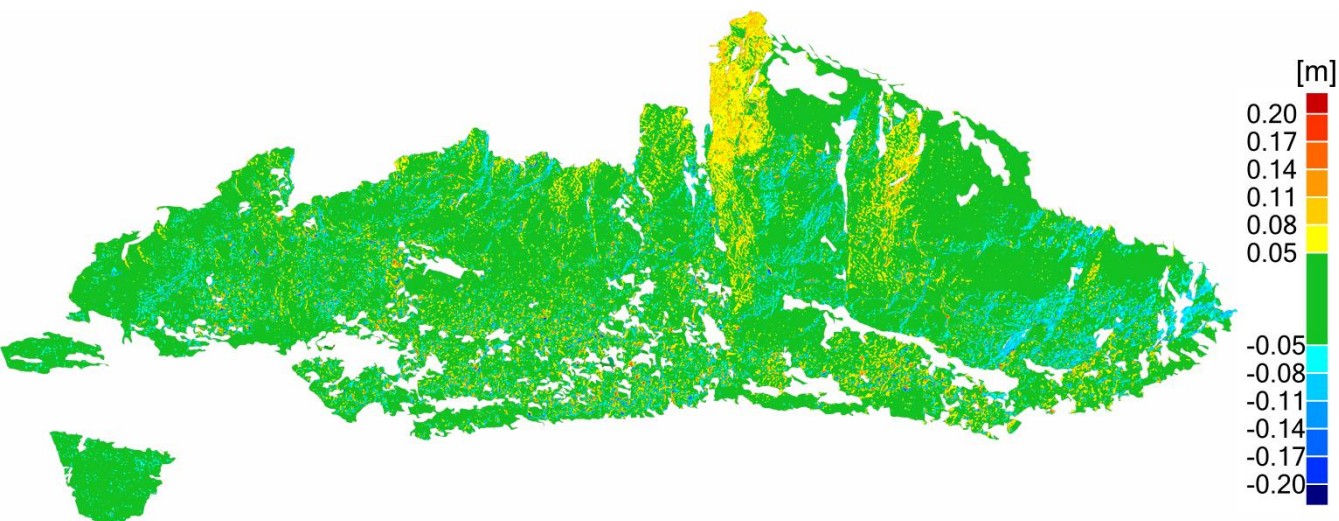

**Figure 13: Comparison between reference dataset and results generated based on the ICProx-algorithm.**

## 5    Summary and outlook

In this contribution an algorithm, termed the iterative closest proximity-algorithm (ICProx) was used to automatically identify and finally exclude deformation from point clouds captured at different points in time. If this diagnostic step is not performed, systematic falsification of the registration is inevitable and consequently wrong conclusions are drawn during analysis of the deformation pattern or computation of geomorphometric measures. Based on the detected stable regions transformation parameters were computed as a vital prerequisite for deformation monitoring. The chosen study areas featured a glacier, a rock glacier, as well as a rockfall area. The datasets themselves proved to be demanding due to individual geometric characteristics, extent, relative amount and/or magnitude of deformations. In general, the algorithm performed well for the data presented in sections 3.1 and 3.2 where geometric changes occurred in a connected fashion. The most challenging task was represented by the data discussed in section 3.3 due to the fact that deformation was scattered all over the datasets and that the deformation rates were comparably low in comparison to the local uncertainty values. All results were numerically and visually evaluated against reference datasets. The mean classification accuracy for stable regions is 68%, while the corresponding measure for deformation is 76%. Thus, the results can be rated as quite promising. However, the importance of carefully acquiring data is particularly apparent in section 4.3 as it clearly demonstrates the inherent link between local sampling density and potential detectability of deformation. It is important to point out that the effects provoked by the sampling uncertainty of TLS notably surpass their capabilities in terms of achievable 3D-precision.

Further research will focus on eliminating some existing disadvantages of the algorithm as well as the applied assessment procedure. One of them is the dependency of the outcome to the degree of decomposition of the point clouds by usage of an octree structure. Hence, an extension has to be implemented that is capable to determine an optimal octree size under

consideration of the local topography. This information could also be used to dismiss certain octree cells due to insufficient geometric properties that would otherwise occur in the algorithm's numerical and visual assessment. In addition, stable cells that border on deformed regions should be split and dismissed from the registration in order to avoid falsification in such transition zones as illustrated in Figure 7 and Figure 9. An alternative strategy could be to apply the ICProx-algorithm in an iterative fashion where the octree size gradually decreases. By this, the boundaries of deformed regions would be detected at a higher resolution. Since geomorphometric monitoring is usually carried out frequently over long periods, strategies have to be developed in order to process time series that consist of several epochs.

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
