# Peer review of "Identification of stable areas in unreferenced laser scans for automated geomorphometric monitoring"

_Earth Surface Dynamics, 2017_

## Referee Comment (RC1) · R.C. Lindenbergh (Referee) · 19 Sep 2017

**Review Report, Esurf-2017-41**

Journal: Earth Surf. Dynamic. Discuss.

Manuscript number: esurf-2017-41

Title: Identification of stable areas in unreferenced laser scans for automated geomorphometric monitoring

**Summary**

For the purpose of topographic monitoring of dynamic sites, the authors propose a method that identifies stable areas and quantifies changes elsewhere.  The method takes unorganized point cloud data as input, as is for example obtained by terrestrial laser scanning. The method builds on a well-known existing method, the so called Iterative Closest Point (ICP) approach. The authors do not apply this method globally over the full area of interest, but divides the area over many small blocks and identify changes by finding those blocs where the local results deviate. The method is demonstrated on three different study areas, all three with different deformation phenomena, one normal glacier, one rock glacier and a landslide area.

**Advice**

Major revision.  The approach the author proposes is appropriate and promising for the type of dynamics considered. The storyline and the discussion should be improved in my opinion before the manuscript is a clear read.  Notably the authors should:

- Make the text more concise: often parts of sentences can be skipped without affecting the contents, and often the order of presented material is not optimal. Some suggestions are given below.
- Reorder some of the material and notably present Ch 2 as related/exiting work
- Check the influence of some parameters on the results: thinned point clouds and various voxel sizes.

**Remarks**

1. Overall: can you characterize in which cases ICProx works better? E.g. when deformation is all scattered over the scene (rock fall?), or when it is more focused, e.g. when considering a glacier that flows between stable rock beds. Related: could you characterize the different movement related/expected for the three geomorphological processes you consider?
2. Overall: (maybe for discussion): what would be a strategy in case of more than two epochs?
3. Overall: could the method be improved by making it iterative (smaller voxels) to better detect the boundaries of deforming areas?

4. Overall: or alternatively: could it be extended by a testing step, that after application of optimal transformation parameters, detect which difference can be considered change, given the quality of the data?

5. Overall: how should the voxel size be chosen given the point density and the expected change?

6. Abstract: -> "E.g. terrestrial laser scanners capture a region of interest in a quasi-laminar …"

7. Abtstract: you mention a central problem, but do not state it explicitly, please do so. "The central problem considered in this work is…" To me this seems to be target free registration in areas where it is not clear where stable subareas are.

8. Abstract: -> "For every case, two epochs… is 70 % on average". Also mention here the reference data

9. Intro: -> "Monitoring surface changes…"

10. Intro, r24: "occurrences", of what? Please specify.

11. P2r5: "congruence model"? Either explain or omit it here.

12. P3r15: explain in 1-2 lines what *radiometric features* are and why these can be used for registration; Same, explain what is meant by *direct georeferencing.*

13. P3r31: -> "A substantial advantage of the last strategy is the actual use of the information present in the point cloud".

14. P4r2: -> "In order to obtain a satisfactory..or pre-alignment algorithms" (skip point clouds)

15. P4r16-r20 and r 22-24: double information: avoid repetitions

16. P4r17-> "and identification of deformation.."

17. P14r18: again *congruent* is mentioned but not explained.

18. Ch 2 + Ch 2.1 (p3) could be called "Existing approaches" as this reviews literature. Your methodology basically starts at Ch 2.2, please make this clear.

19. P4r29: "The first one automatically….data. For this purpose the 4PCS-algorithm is…

20. P4, last paragraph: what is the third step of Phase 1?

21. P5r4: -> "that occurred between scan acquisitions"

22. P5r13-r14: "to clarify…object's surface": this I don't get, please clarify.

23. P5r15: how are these "stable regions" obtained?

24. P5r17: -> "deformation estimation" (not measurement)

25. P5r20: -> "is to identify"

26. P5r21: I would not call this "segmentation" but "decomposition" or "voxelization"

27. Table 1: make separate columns for GSD and scanning increment

28. P7r5: "…at an elevation…and is ~ 900 m long and ~350 m wide"

29. P7r11: "and has reported to have a slower upper part…"

30. P7: last paragraph: this is discussing existing work and fits better in a Ch 2 "Existing approaches"

31. Fig.3, caption I see a white circle, not a red circle?

32. P8r9: -> "in the Central part of the Austrian Alps"

33. P8r15: -> "The current retreat of.. glacier has been observed using TLS since 2001… while 6.3m vertical loss was detected…"

34. P9r11: "computed transformation parameters" computed by who?

35. P10: "nunatak mountains"??? Please explain

36. Ch 4: p11: this page contains more general info, (like classification problem). This kind of info should be presented earlier, chapter 1, or chapter 3. Here just present and analyze the results.
37. Also here you could mention once how the reference data is obtained; Later in the discussion: what is the effect of the location of the selected stable areas? If this location is rather small or badly covering the full extent of the ROI, this could affect the validation
38. Ch4: what do you mean by "inspection map"? Is there not a more descriptive name for these images? Is this not just a "change map"?
39. P14 r14: -> "in orange corresponding to large and detected geometric changes"
40. P14r24: can you quantify these "margin of detectability"?
41. P18: could you do and add some experiments with different resamplings (thinning of the data set)?

---

## Referee Comment (RC2) · Anonymous Referee #2 · 22 Sep 2017

The paper "Identification of stable areas in unreferenced laser scans for automated geomorphometric monitoring" presents a useful approach in the context of multitemporal analysis of laser scanner points clouds for change detection. The paper has good potentialities and the approach seems promising, however there are several critical points requiring major revision. As a first point the paper is not easy readable, the structure should be improved and the English language checked. In regard to the methodology and the presented case studies there are various points requiring a better explanation or even further analysis. Some of the more important critical points are listed in the following lines; for more technical comments see attached pdf (commented).

[Figure]

1) The methodology is not clearly explained, including its limitations: for example how is defined the dimension of octree cells? Moreover, for a good alignment, how the stable areas should be spatially distributed in the studied domain? In your figure 1 the "stable cubes" are almost all clustered in the corner.

2) The most critical point of the article for me is the validation of the proposed approach. As first point, it could be worth to test your approach with synthetic data sets (I mean simulated) in which you can control noise, location of stable areas, the distortion between point clouds, etc.. Clearly, reality is generally more complex; however, with synthetic data at least you are sure of how the algorithm works in defined conditions. Then, the validation of your approach for the identification of stable and unstable areas in front of the reference data sets seems to present some weakness; from your discussion is not clear how much accurate are your reference data sets. Linked to this, from your description of ground control points for the different study sites, the reader may draw the conclusion that it would be difficult to define stable and unstable areas with comparative approaches: so, one can expect high uncertainty in your reference data sets. Maybe you could use a case study in which you have a more efficient network of control points so as to have a more accurate and precise reference dataset for comparison.

3) In the section 5 "summary and outlook" you give mean values and standard deviations for classification of stable and unstable areas; I think that calculating mean values and standard deviations from 3 samples is not much informative.

Please also note the supplement to this comment:
https://www.earth-surf-dynam-discuss.net/esurf-2017-41/esurf-2017-41-RC2-supplement.pdf

**Supplement:**

[revised manuscript text omitted]

---

## Referee Comment (RC3) · Anonymous Referee #3 · 21 Oct 2017

This manuscript aimed to use ICProx algorithm to determine deformations that occurred at a rock glacier. It is an interesting research, but it has a far distance to acceptance. Some of my advices are the following:

(1) The mehtod is presented by this reserach or by other researches. It is not very clear.

(2) In abstract, the content about this research is limited. Most of the contents are background introduction.

(3) From the introduction part, I can not found what this research aims to do and why conduct this research?

[Figure]

(4) The manuscript does not have "Discussion" part. Then, what about this research compared to relevant researches?

(5) The accuracy is very low in the first study area, especially for the deformation area.

Overall, I suggest the authors revise the manuscript in written and structure aspects. I do not think the manuscript can be accepted at its current form.

---

## Author Comment (AC2) · 1 Dec 2017

Dear colleague,

Thank you very much for your effort and valuable comments. We have tried to consider every single one of your suggestions – hopefully to your satisfaction.

Best regards

The Authors

**Response to reviewer #1**

1. Overall: can you characterize in which cases ICProx works better? E.g. when deformation is all scattered over the scene (rock fall?), or when it is more focused, e.g. when considering a glacier that flows between stable rock beds.

We have added some sentences to the conclusion: "In general, the algorithm performed well for the data presented in sections **Error! Reference source not found.** and **Error! Reference source not found.** where geometric changes occurred in a connected fashion. The most challenging task was represented by the data presented in section **Error! Reference source not found.** due to the fact that deformation was scattered all over the datasets and that the deformation rates were comparably low.

2. Related: could you characterize the different movement related/expected for the three geomorphological processes you consider?

It would be possible to compute deformation vectors similar to Teza et al. (2007) after the datasets would have been registered. However, the approach would only lead to one vector per octree cell and would hence be a "mixed signal" meaning that parts of the cell could move into different directions in reality. Hence, we're currently working on entity-based approaches rather than generating numbers for individual voxels.

Teza, G., Galgaro, A., Zaltron, N., Genevois, R. (2007): Terrestrial laser scanner to detect landslide displacement fields: a new approach. International Journal of Remote Sensing, 28(16), pp. 3425-3446.

3. Overall: (maybe for discussion): what would be a strategy in case of more than two epochs?

We have added one sentence to the outlook section. However, a general strategy cannot be mentioned since every area of interest requires its own unique strategy, e.g. register all epochs into the reference epoch or register pairwise, epoch 1 into reference epoch, epoch 2 into the geodetic datum of epoch 1 and so on.

"Since geomorphometric monitoring is usually carried out frequently over long periods, strategies have to be developed in order to process time series that consist of several epochs."

4. Overall: could the method be improved by making it iterative (smaller voxels) to better detect the boundaries of deforming areas?

The reviewer's suggestion has been considered. The following sentences have been added:

"An alternative strategy could be to apply the ICProx-algorithm in an iterative fashion where the octree size gradually decreases. By this, the boundaries of deformed regions would be detected at a higher resolution."

5. Overall: or alternatively: could it be extended by a testing step, that after application of optimal transformation parameters, detect which difference can be considered change, given the quality of the data?

This is in fact one of the key features of the algorithm and is hence already implemented. Please see Wujanz et al. (2016) from page 8 onwards for details.

Wujanz, D., Krueger, D. and Neitzel, F.: *Identification of Stable Areas in Unreferenced Laser Scans for Deformation Measurement*. The Photogrammetric Record, 31(155), pp. 261-280, 2016.

6. Overall: how should the voxel size be chosen given the point density and the expected change?

These are two very different aspects. The voxel size controls the resolution of the result – the smaller it is, the finer will be the resolution of the classification process. The local point density on the other hand, which is an inherent characteristic of the input data, controls the detectability. So, if the local point sampling is low, the local detectability of deformation is also comparably low. Quite frank, the algorithm cannot do anything about the data quality. We have mentioned this aspect in the conclusion.

"However, the importance of carefully acquiring data is particularly apparent in section **Error! Reference source not found.** as it clearly demonstrates the inherent link between local sampling density and potential detectability of deformation. It is important to point out that the effects provoked by the sampling uncertainty of TLS notably surpass their capabilities in terms of achievable 3D-precision."

We have also added two sentences to section 3.

"As already shown in Wujanz et al. (2016b) the choice of the octree cell size has notable effects onto the outcome. In general, a smaller octree size leads to a higher degree of detecting stable regions while the computational effort notably increases. Hence, setting the octree size is a balancing act between resolution and computation time."

6. Abstract: -> "E.g. terrestrial laser scanners capture a region of interest in a quasi-laminar …"

The sentence has been rephrased to "…e.g. by capturing a region of interest with terrestrial laser scanners which results in a so-called 3D point cloud."

7. Abstract: you mention a central problem, but do not state it explicitly, please do so. "The central problem considered in this work is…" To me this seems to be target free registration in areas where it is not clear where stable subareas are.

We tried to argue in a sequential fashion: Problem #1: Computation of transformation parameters. Problem #2: How can one achieve this and what are possible disadvantages? Etc.. In order to clarify this issue the sentence has been altered to:

"The central problem in deformation monitoring is the transformation of 3D point clouds captured at different points in time (epochs) into a stable reference coordinate system."

Abstract: -> "For every case, two epochs… is 70 % on average". Also mention here the reference data

Sentence has been altered to:

"For every example two epochs were processed while the ICProx-algorithm's classification accuracy is 70% on average in comparison to reference data."

9. Intro: -> "Monitoring surface changes…"

The authors have addressed the reviewer's remark.

"Monitoring surface changes in hazardous areas has been an important task of the geodetic community in the last decades."

10. Intro, r24: "occurrences", of what? Please specify.

This sentence now reads:

"Due to the predicted increase of natural disasters (Anderson and Bausch 2006) this problem domain will hence gain importance in the future."

11. P2r5: "congruence model"? Either explain or omit it here.

Parts of the sentence have been deleted. "In order to process the acquired TLS data, an appropriate deformation model has to be chosen (Heunecke and Welsch 2000)."

12. P3r15: explain in 1-2 lines what radiometric features are and why these can be used for registration;

The according part has been extensively extended and now reads:

"Point clouds captured by terrestrial scanners also contain radiometric information (Höfle and Pfeifer 2007), which is referred to as intensity, in addition to the geometric content. Intensity values are assigned to individual points and are based on the strength of a reflected signal. If the topology of points within a dataset is known this information can be used to convert a point cloud into an image where intensity values represent the brightness of individual pixels. By doing this, well established techniques from the field of image matching can be applied to 3D-datasets. As a first step so called keypoints have to be extracted, which are distinct features within a local neighbourhood in terms of their grey scale values. After keypoint extraction descriptors are used to establish correspondences between keypoints from different datasets (Lowe 1999). Based on this information transformation parameters can be computed. Since the long-term stability of radiometric information captured by TLS has not yet been studied, this strategy is not considered in the following. The successful application over short periods of time has been demonstrated by Böhm and Becker (2007)."

Same, explain what is meant by direct georeferencing.

The according sentence has been extended to "Instead of computing transformation parameters based on established correspondences the relation between different local coordinate systems can also be achieved by observing their location and orientation within a superior coordinate frame. For this, additional sensors are required while the general concept is well known, for instance in aerial photogrammetry (Cramer et al. 2000). Methods for direct georeferencing of TLS were initially just of scientific interest (e.g. Paffenholz et al. 2010) while lately several commercial systems emerged (Riegl 2017), (Zoller + Fröhlich 2017). A significant drawback of direct georeferencing is the extension of a scanner's error budget (Soudarissanane 2016), due to the use of additional positioning- and orientation sensors such as GNSS-equipment or electronic compasses. With increasing scanning range the impact onto the relative rotation between two point clouds also increases, which is a result of the limited accuracy of compasses."

P3r31: -> "A substantial advantage of the last strategy is the actual use of the information present in the point cloud".

The reviewer's suggestion has been considered.

14. P4r2: -> "In order to obtain a satisfactory..or pre-a lignment algorithms" (skip point clouds)

The reviewer's suggestion has been considered.

"In order to obtain a satisfactory pre-alignment, several strategies appear to be suitable such as direct georeferencing, manual pre-alignment or pre-alignment algorithms."

15. P4r16-r20 and r 22-24: double information: avoid repetitions

The reviewer's suggestion has been considered.

"In order to avoid falsification of the deformation measurement process and consequently the analysis of deformation, it is unavoidable to identify deformed areas within point clouds and to reject them from the

computation of transformation parameters. For this task the iterative closest proximity-algorithm (ICProx) was used that consists of three essential phases:"

16. P4r17-> "and identification of deformation.."

The reviewer's suggestion has been considered.

17. P14r18: again congruent is mentioned but not explained.

The reviewer's suggestion has been considered.

"Congruent respectively stable areas in terms of geometry are detected by a combinatorial approach termed the maximum subsample method (Neitzel 2004) which is, in terms of robustness against outliers/deformation, more reliable and hence superior to robust estimators or RANSAC."

18. Ch 2 + Ch 2.1 (p3) could be called "Existing approaches" as this reviews literature. Your methodology basically starts at Ch 2.2, please make this clear.

The structure of chapter 2 has been altered. The caption of Ch. 2.1 has been removed while the content of 2.2 was moved to chapter 3.

P4r29: "The first one automatically....data. For this purpose the 4PCS-algorithm is...

Removal of the first part of the sentence as suggested by the reviewer may confuse the reader since the third sentence in this paragraphs refers to "a step" and not a "phase". Hence, the sentence has not been modified.

20. P4, last paragraph: what is the third step of Phase 1?

The reviewer's suggestion has been considered.

"Subsequently the ICP-algorithm is locally applied within each octree cell that represents the last step of Phase 1."

21. P5r4: -> "that occurred between scan acquisitions"

The reviewer's suggestion has been considered.

"The general idea behind this approach is to locally increase or decrease the portion of deformation in order to identify geometric changes that occurred between scan acquisitions by means of suitable criteria in Phase 2."

22. P5r13-r14: "to clarify...object's surface": this I don't get, please clarify.

The reviewer's suggestion has been considered.

"This information is required in order to determine if an octree cell can be associated to a congruent set of cells or not."

"The general concept of this feature considers the fact that every scan of a stable object yields in a different point sampling (Wujanz et al. 2016b)."

23. P5r15: how are these "stable regions" obtained?

The reviewer's suggestion has been considered.

"This time however only stable regions, that were detected in Phase 2, serve as input which are processed as a whole and not in segments."

24. P5r17: -> "deformation estimation" (not measurement)

The reviewer's suggestion has been considered.

The result of Phase 3 is a set of transformation parameters that forms the basis for the deformation estimation.

25. P5r20: -> "is to identify"

The reviewer's suggestion has been considered.

26.P5r21: I would not call this "segmentation" but "decomposition" or "voxelization"

The reviewer's suggestion has been considered.

27. Table 1: make separate columns for GSD and scanning increment

The reviewer's suggestion has been considered.

28. P7r5: "...at an elevation...and is ~ 900 m long and ~350 m wide"

The reviewer's suggestion has been considered.

29. P7r11: "and has reported to have a slower upper part..."

The reviewer's suggestion has been considered. The according sentence was quite lengthy and thus has been split into two sentences.

"The rock glacier itself shows distinct patterns of surface elevation changes and surface displacements. In addition, the rock glacier has reported to have a slower upper part and a rapidly moving lower part with maximum horizontal displacements of up to 2.5 m per year (Avian et al. 2009)."

30. P7: last paragraph: this is discussing existing work and fits better in a Ch 2 "Existing approaches"

The authors tried to move the very first sentence of the last paragraph into chapter 1 or 2 yet it did not really match the given content. Hence, the sentence remains as it was. The remaining references refer to research that is directly connected to the observed site and thus also remained in its original form.

31. Fig.3, caption I see a white circle, not a red circle?

The reviewer's suggestion has been considered.

"Figure 3: Scanner configuration at Hinteres Langtalkar rock glacier. The scanning position HLK is indicated by a red ellipse with a white edge, the rock glacier outline by a white dashed line. White circles with green edges highlight registration points (image, 28.08.2012)."

32. P8r9: -> "in the Central part of the Austrian Alps"

The suggestion by the reviewer has been considered.

"The Pasterze glacier (N 47°04', E 12°44') is located in the Central part of the Austrian Alps."

33.P8r15: -> "The current retreat of.. glacier has been  observed using TLS since 2001...

"The current retreat of the terminus area of Pasterze glacier has been observed using TLS since 2001 covering an area of appr. 0.9 km²."

while 6.3m vertical loss was detected..."

The reviewer's suggestion has been considered.

"On average the annual elevation change within the debris covered part sums up to approximately 3.7 m while 6.3 m vertical loss was detected in the clean ice section between 2011 and 2012."

34. P9r11: "computed transformation parameters" computed by who?

The reviewer's suggestion has been considered.

"As a consequence the computed transformation parameters based on the reference points can only be seen as an approximate solution as they are very likely to be subject of extrapolative effects and hence have been refined by surface based registration."

35. P10: "nunatak mountains"??? Please explain

The reviewer's suggestion has been considered. An additional reference has been added that covers this geological term.

"The rock fall area Burgstall comprises the two former nunatak mountains (Kellerer-Pirklbauer et al. 2012) of Mittlerer and Hoher Burgstall which were encompassed from Pasterze glacier until the beginning of the 20th century."

36. Ch 4: p11: this page contains more general info, (like classification problem). This kind of info should be presented earlier, chapter 1, or chapter 3. Here just present and analyze the results.

We have carefully considered your suggestion and moved the according paragraph into different chapters. However, we decided to move back to the original logical sequence since the described assessment scheme is quite important for the reader since the results are presented in the very same section.

37. Also here you could mention once how the reference data is obtained; Later in the discussion: what is the effect of the location of the selected stable areas? If this location is rather small or badly covering the full extent of the ROI, this could affect the validation.

We have mentioned how reference data was generated in every subsection of chapter 4, for instance (from section 4.3): "Yet again, the sampling of reference points has to be rated as unfavourable. Hence, initial transformation parameters have been computed based on the reference points. Afterwards, deformed regions were manually rejected while the remaining points were used to refine the relative alignment between points clouds based on the ICP."

38. Ch4: what do you mean by "inspection map"? Is there not a more descriptive name for these images? Is this not just a "change map"?

The reviewer's suggestion has been considered. The problem is that there are numerous names for the very same outcome such as inspection map, difference map, change map, deformation map,…. We decided to replace "inspection" with "change"

39. P14 r14: -> "in orange corresponding to large and detected geometric changes"

The reviewer's suggestion has been considered.

"…glacier tongue is highlighted in orange corresponding to large detected geometric changes."

40. P14r24: can you quantify these "margin of detectability"?

Not exactly, since we cannot visualise the uncertainty of distances between octree cells (the illustration would be chaotic while many coloured lines are overlapping). Instead we have added additional figures where the uncertainty values of individual octree cells are depicted. This shows the reader the heterogeneous distribution and magnitude of uncertainties.

41. P18: could you do and add some experiments with different resamplings (thinning of the data set)?

After long discussions we have decided not to reprocess the data even though we have done similar experiments at the very beginning of this project. We believe that the effect of sampling uncertainty is now a lot clearer to the reader since we have added some more figures according to comment #40.

---

## Author Comment (AC3) · 1 Dec 2017

Dear colleague,

Thank you very much for your effort and valuable comments. We have tried to consider every single one of your suggestions – hopefully to your satisfaction.

Best regards

The Authors

**Response to reviewer #2**

Comment 1:    I would expand a little this part, maybe you could also refer directly with the term "intensity".

The reviewer's suggestion has been considered. Please see comment 12. By reviewer #1

Comment 2:    Expand, furnish more information on the methodology.

The reviewer's suggestion has been considered.

"As a first step planar segments have to be extracted from the original point clouds. Then, identical planes are computed instead of matching single points such as in radiometric approaches. By using planes instead of points, the precision of the resulting transformation parameters notably increases. However, the approach relies on the existence planar areas within a region of interest and is hence mostly applied in urban environments. The most popular registration method uses redundantly captured regions of two point clouds and is called the iterative closest point algorithm (ICP) as proposed by Besl & McKay (1992). A substantial advantage of the last strategy over the aforementioned ones is the actual use of the information present in the point cloud. A drawback of the algorithm is its dependence to a sufficient pre-alignment of two datasets."

Comment 3:    I'm not sure that this subsection title is fitted to the matter. Maybe simply "pre-alignment methodology" or something similar is more effective.

The reviewer's suggestion has been considered by restructuring the manuscript. The mentioned paragraph can now be found under section 2 existing methods.

Comment 4:    But this could be in contradiction with the need to avoid "any form of user-interaction", stated above. At least this could be the first impression the reader could have; however if you say explicitly that ransac method automatically filters out sensitive areas (I mean the ones with high deformation leading to problems of alignment) the possible contradiction is avoided.

The reviewer's suggestion has been considered.

"Kromer et al. (2017) approach this issue by applying Fischler and Bolles' (1981) Random sample consensus (RANSAC) in order to **automatically** reject correspondences between two epochs which either arise as a consequence of deformation or false matches."

In addition, the following sentence has been modified:

"In order to avoid falsification of the deformation measurement process and consequently the analysis of deformation, it is unavoidable to **automatically** identify deformed areas within point clouds and to reject them from the computation of transformation parameters"

Comment 5:    Which is the meaning of "object space"???

The reviewer's suggestion has been considered. The term object space has been replaced by "area of interest".

The reviewer's suggestion has been considered by rephrasing the sentence. " The ICProx is based on a spatial segmentation of the original datasets and identifies deformation respectively stable areas by comparison of locally computed transformation parameters for individual segments (Wujanz et al. 2016b). "

I will fuse this section with 2.1

The reviewer's suggestion has been considered by restructuring the according sections.

Comment 7:     Say some words to give information on the approach. Sentence not much clear, explain better. Consider rephrasing, the link with the preceding sentence is not clear.

The reviewer's suggestion has been considered.

"Phase 1 in turn contains **three steps** while the first one automatically carries out the pre-alignment of data. Therefore the 4PCS-algorithm as proposed by Aiger et al. (2008) is used in the **first step**. In the **second step**, the coarsely aligned data is segmented in cubes of equal size that are also referred to as octree cells (Samet 2006 p. 211 ff). Subsequently the ICP-algorithm is locally applied within each octree cell that represents the **last step** of Phase 1.

In the current implementation this step is based on empirical settings. We have mentioned the following in the outlook: "Hence, an extension has to be implemented that is capable to determine an optimal octree size under consideration of the local topography."

Comment 9:       Are the candidates stable areas in the selected subdomains???

No, at the very beginning it is not clear which regions in the point clouds are stable. It is the task of the ICProx-algorithm to clarify this issue. Candidates are samples that reduce the data volume of the original dataset. This step is carried out in every implementation.

Comment 10:       Not clear, is the portion of deformation the size of cubes (octree cells)?

No, but the size of the cubes influences the local amount of deformation. The smaller the cubes the higher the local influence of deformed / stable areas. If the octree size is too small, the ICP will simply fail since it will converge into local minima.

The sentence has been rephrased. See also comment 22 of reviewer #1.

"The general concept of this feature considers the fact that every scan of a stable object yields in a different point sampling (Wujanz et al. 2016b)."

Comment 12:       I'm wondering if the stables areas should have some requirements (e.g., relative amount of stable areas over unstable, geographical distribution, etc.) in order to have a final good alignment. From your figure 1 stable areas is almost all located on the left side...

The reviewer's suggestion has been considered.

"From a geodetic perspective the most desirable arrangement would be a more or less connected region that is subject to deformation which is surrounded by stable areas. In addition, the ratio between stable and deformed regions should be rather large in order to being able to compute transformation parameters with the highest possible redundancy. However, the distribution of stable areas as well as their relative amount are usually unpredictable in practice since every region of interest has got its own individual characteristics. Countermeasures in order to receive favourable arrangements can be achieved by carrying out the surveys more frequently and/or to perform panorama scans which increases the chance of capturing additional stable areas. It is obvious that the result depicted on the right of **Error! Reference source not found.** may yield in imprecise transformation parameters due to the fact that a large amount of octree cells are subject to deformation while the arrangement of stable cells is unfavourable."

Comment 13:     I would use letters to identify figures.

After discussing your suggestion we have decided to stick to our original scheme.

Comment 14:     Did you tried the suggested approach on synthetic data so as to have a kind of benchmark?

We did not use synthetic data yet we have used two different scenarios in order to verify the approach (see Wujanz et al. 2016). In the first one a scanner's viewpoint remained constant over the course of several captured epochs. By this, reference transformation parameters were known: they had to be all 0! In the second one the deformation was locally restricted and well defined (we have used data from a crash test). A third alternative strategy was used in the submitted manuscript.

Wujanz, D., Krueger, D. and Neitzel, F.: *Identification of Stable Areas in Unreferenced Laser Scans for Deformation Measurement*. The Photogrammetric Record, 31(155), pp. 261-280, 2016.

Comment 15:     Are you sure that this manually classified map correspond to the true?  How you identified stable and unstable areas????

Every manually generated ground truth contains subjective influences. In order to overcome this, we generated the reference in an iterative fashion.

1. Register point clouds based on ICP
2. Generate change map
3. Remove areas with systematic shift / deformation
4. Repeat steps 1 to 3 until only random noise remains

Please also have a look at our remarks to comment 20

Comment 16:     "….were sampled with 0.75 m on average."

Every implementation of the ICP contains a downsampling step that reduces the input data in order to i) reduce the required effort for correspondence determination and ii) to restrict the size of the normal equation matrix during parameter estimation. A disadvantage of randomly sampling these "candidates" is that the majority of samples will be located closer to the scanner since the local point density is also closer opposed to regions of the point cloud that are further away. The defined measure means that one candidate should be located every ~0.75 m. This ensures a (approximately) systematic sampling over the original point cloud.

Comment 17:     Explain better how the georeferenced reference has been derived as well as its accuracy.

The location of all applied tie points are highlighted in Figure 3 to 5. From a geodetic point of view it is not meaningful to report their accuracy as the distribution is quite unfavourable. One could have placed all tie point 1 m away from the scanner which would yield in very small residuals yet, these values are only valid for this very limited range. We have critically commented this issue throughout the contribution.

Section 4.1 "It can notably be seen that the distribution is rather unfavourable from a geodetic point of view as the active zone of the glacier, that is highlighted by a white dashed line, is not surrounded by registration points. This may lead to extrapolative respectively leverage effects."

Section 4.2 "As a consequence the computed transformation parameters based on the reference points can only be seen as an approximate solution as they are very likely to be subject of extrapolative effects and hence have to be refined by e.g. surface based registration."

Comment 18:     Insert legend for colour coded map.

We decided to leave the written description of the colour map at the very beginning of chapter 5. The information was to lengthy for the figure.

Comment 19:     Quite high standard deviation, even if I don't think that with 3 values you can compute confidently a standard deviation....I would say that the accuracy of prediction of unstable areas is quite variable (worst case table2).

We have removed the listed standard deviations. The accuracy of the algorithm of course depends on the geometric characteristics of the area of interest, the chosen octree size as well as the point density of the acquired datasets. We have mentioned this aspect several times throughout the contribution, e.g.

"Hence, an extension has to be implemented that is capable to determine an optimal octree size under consideration of the local topography. This information could also be used to dismiss certain octree cells due to insufficient geometric properties **that would otherwise occur in the algorithm's numerical and visual assessment**."

The worst example in this contribution is listed in table 3 (13% of undetected deformation) and not in table 2 (2% of undetected deformation). We have also listed reasons for this effect:

"The numerical assessment of the generated results is depicted in **Error! Reference source not found.** and shows that a comparably large part of deformed areas was not correctly assigned as indicated by the numbers in the red coloured cell. The reason for this is twofold and can be found both in the ratio among the low magnitude of deformation and the corresponding local spatial uncertainties as well as the heterogeneous occurrence of geometric changes."

Comment 20:     To feel more confident I would try the method on sintethic data set where you know truly which are stable and unstable areas. As a second option you should use a case study in which you have a more efficient network of control points so as to have a more accurate and precise reference dataset for comparision.

The precision of the reference data is of course dependent to the registration. We have shown three strategies of how to deal with this problem; please see remarks to comments #14 and 15. Yet, what should also be considered is the influence of the deformation pattern: the chosen settings of these colour coded representations have a large impact onto the outcome and hence, the confusion maps that we produce based on it. If you ask three people to generate change maps you'll most likely end up with three very different results as shown below.

[Figure]

Hence, we have implemented an algorithm that visualises only statistically significant deformations in dependence to the:

- Local sampling density (as we've done in the submitted manuscript)
- Stochastic properties of the applied scanner (Wujanz et al. 2017)
- Inhomogeneous influence of the registration, in contrast to Lague et al. (2013)

Lague, D., Brodu, N. and Leroux, J.: *Accurate 3D comparison of complex topography with terrestrial laser scanner: application to the Rangitikei canyon (NZ)*. ISPRS Journal of Photogrammetry and Remote Sensing, 82, pp. 10-26, 2013.

Wujanz, D., Burger, M., Mettenleiter, M. and Neitzel, F: *An intensity-based stochastic model for terrestrial laser scanners*. ISPRS Journal of Photogrammetry and Remote Sensing, vol. 125, pp. 146-155, 2017.

---

## Author Comment (AC4) · 1 Dec 2017

Dear colleague,

Thank you very much for your effort and valuable comments. We have tried to consider every single one of your suggestions – hopefully to your satisfaction.

Best regards

The Authors

**Response to reviewer #3**

This manuscript aimed to use ICProx algorithm to determine deformations that occurred at a rock glacier. It is an interesting research, but it has a far distance to acceptance. Some of my advices are the following:

(1) The mehtod is presented by this reserach or by other researches. It is not very clear.

We were not sure what you meant by this.

(2) In abstract, the content about this research is limited. Most of the contents are background introduction.

In general, we as Geodesists try to answer the following questions in the abstract:

What is the general problem?

"Current research questions in the field of geomorphology focus on the impact of climate change on several processes causing subsequently natural hazards. Geodetic deformation measurements are a suitable tool to document such geomorphic mechanisms e.g. by capturing a region of interest with terrestrial laser scanners which results in a so-called 3D point cloud."

What is the particular problem?

"The central problem in deformation monitoring is the transformation of 3D point clouds captured at different points in time (epochs) into a stable reference coordinate system. To date, this step has been mostly carried out by usage of artificial targets and/or control points. Several drawbacks are related to this strategy such as the enormous effort to distribute the targets within the area of interest, the required survey by additional geodetic sensors such as total stations or GNSS-receivers as well as the limited extent within the region of interest."

How do we solve the particular problem?

"In this contribution a surface-based registration methodology, termed the iterative closest proximity algorithm (ICProx), that solely uses points as input, similar to the iterative closest point-algorithm (ICP), and hence does not require any artificial targets or extracted geometric primitives, such as planes. The aim of this study was to automatically classify deformations that occurred at a rock glacier, an ice glacier as well as in a rock fall area. For every example two epochs were processed while the ICProx-algorithm's classification accuracy is 70% on average in comparison to reference data."

(3) From the introduction part, I can not found what this research aims to do and why conduct this research?

At the end of the first chapter we state the following: "The most delicate step in this processing chain (linked to deformation monitoring) is the transformation of single epochs into a common reference coordinate system, which is also referred to as registration or matching of point clouds. Erroneous effects that occur in this step have an immediate and systematic impact onto the quantification of deformation. Thus, all conclusions that are drawn based on the generated results are falsified."

(4) The manuscript does not have "Discussion" part. Then, what about this research compared to relevant researches?

We have structured the manuscript in three major parts:

- Section 3: Description of the algorithm
- Section 4: Description of the areas of interest
- Section 5: Discussion of the results

Regarding other research: There is no other algorithm than ours that is capable to automatically identify deformation within unregistered point clouds. Hence, we could not compare to other implementations.

(5) The accuracy is very low in the first study area, especially for the deformation area. Overall, I suggest the authors revise the manuscript in written and structure aspects. I do not think the manuscript can be accepted at its current form.

We weren't sure if you were referring to the classification accuracy. If so, we have added the following sentence that explains the cause of this result.

"Hence, an extension has to be implemented that is capable to determine an optimal octree size under consideration of the local topography. This information could also be used to dismiss certain octree cells due to insufficient geometric properties that would **otherwise occur in the algorithm's numerical and visual assessment**."

---

## Author Comment (AC5) · 1 Dec 2017

Please see attached supplement

Please also note the supplement to this comment:
https://www.earth-surf-dynam-discuss.net/esurf-2017-41/esurf-2017-41-AC5-supplement.pdf